METHODS AND RESOURCES

# Experimental–numerical method for calculating bending moments in swimming fish shows that fish larvae control undulatory swimming with simple actuation

Cees J. Voesenek[1], Gen Li[2], Florian T. Muijres[1], Johan L. van Leeuwen[1]*

**1** Experimental Zoology Group, Department of Animal Sciences, Wageningen University, Wageningen, the Netherlands, **2** Department of Mathematical Science and Advanced Technology, Japan Agency for Marine-Earth Science and Technology (JAMSTEC), Yokohama, Japan

* johan.vanleeuwen@wur.nl

**Data Availability Statement:** All reconstructed kinematics and bending moments as well as source codes for the custom programs used in this

## Abstract

Most fish swim with body undulations that result from fluid–structure interactions between the fish's internal tissues and the surrounding water. Gaining insight into these complex fluid–structure interactions is essential to understand how fish swim. To this end, we developed a dedicated experimental–numerical inverse dynamics approach to calculate the lateral bending moment distributions for a large-amplitude undulatory swimmer that moves freely in three-dimensional space. We combined automated motion tracking from multiple synchronised high-speed video sequences, computation of fluid dynamic stresses on the swimmer's body from computational fluid dynamics, and bending moment calculations using these stresses as input for a novel beam model of the body. The bending moment, which represent the system's net actuation, varies over time and along the fish's central axis due to muscle actions, passive tissues, inertia, and fluid dynamics. Our three-dimensional analysis of 113 swimming events of zebrafish larvae ranging in age from 3 to 12 days after fertilisation shows that these bending moment patterns are not only relatively simple but also strikingly similar throughout early development and from fast starts to periodic swimming. This suggests that fish larvae may produce and adjust swimming movements relatively simply, yet effectively, while restructuring their neuromuscular control system throughout their rapid development.

## Introduction

Swimming is a vital component of the fitness of a fish because fish swim to search for food, hunt prey, escape from predators, migrate and disperse, and manoeuvre through complex environments. Many fish species swim by performing body undulations that result from an interaction between body tissues and the surrounding water [1,2]. Understanding these complex fluid–structure interactions is crucial to gain insight into the mechanics and control of fish swimming [3].

article are available in a Dryad data repository (https://doi.org/10.5061/dryad.2280gb5p6). The source high-speed video data and computational fluid dynamics results are available upon request from the corresponding author.

**Funding:** This work was partly supported by NWO/ALW grant 824.15.001 to JLvL, NWO/VENI-863-14-007 to FTM (Dutch Research Council, NWO; https://www.nwo.nl/en), and the Japan Society for the Promotion of Science (JP17K17641 to GL). The funders had no role in study design, data collection and analysis, decision to publish, or preparation of the manuscript.

**Competing interests:** The authors have declared that no competing interests exist.

To analyse the fluid–structure interactions during swimming, we need to understand the external fluid mechanics (water), the internal solid mechanics (skin, muscle, skeleton), and their coupling. During swimming, the body of the fish moves through the water, inducing a flow around it [4,5]. The resulting fluid dynamic forces interact with the body tissues via the skin, resulting in a change in deformation. This deformation will change the motion of the surface of the body, which influences the fluid dynamic forces, thus forming a loop of tight coupling between the fluid mechanics and the internal solid mechanics [2,6,7]. This complex two-way fluid–structure interaction creates the typical travelling wave pattern observed in swimming fish [8,9].

To better understand how fish swim, we need insight into the internal mechanics of the axial muscles and passive tissues that actuate the motion. Muscle activation patterns can be measured directly with electromyography, in which electrodes are inserted in the muscles to measure potential differences [10–12]. However, this technique may incur considerable changes in swimming behaviour. This holds especially for research on fish larvae; electromyographical experiments require the larvae to be paralysed [13] or fixed in place [14], thus changing the fluid–structure interactions that produce the body wave [15]. Furthermore, the resolution along the body is limited by the number of inserted electrodes.

An alternative is an inverse dynamics approach [16] in which net forces and moments are calculated from measured kinematics. Hess and Videler [17] used a simplified small-amplitude fluid [18] and internal body model to estimate bending moments along the central axis of saithe from the motion of its centreline. The bending moment is defined for each transversal slice along the body as the sum of the moments produced by the muscles and passive tissues, counteracting the moments due to inertia and surrounding water [17,19,20]. Because the muscles are the only component in the system that produces net positive work over a cycle, bending moment distributions are the net actuation of the system and hint toward properties of the muscle activation patterns. Previous pioneering studies [17,19] used a basic small-amplitude beam model and simplified fluid dynamics to compute bending moments for only a few cases of periodic swimming in the inertial regime. However, swimming fish often use large undulatory amplitudes, and smaller fish or fish larvae may swim in the intermediate Reynolds regime, in which inviscid fluid models are invalid. In addition, for many fish species, the majority of swimming (e.g., starts, turns, feeding strikes) cannot be analysed under periodic assumptions. In our newly developed method, we therefore removed these simplifying assumptions. We used three-dimensional reconstructed kinematics [21]; beam theory supporting aperiodic, arbitrarily large deformation amplitudes; and full numerical solutions of the incompressible Navier–Stokes equations to calculate fluid dynamic forces (Fig 1).

To demonstrate this method, we examined the swimming performance of fish larvae. Zebrafish (*Danio rerio*, Hamilton 1822) larvae, the subject of this study, can swim immediately after hatching at considerable speed and tail-beat frequency [3,22,23]. In the next days of development, their bodies change considerably, both externally and internally [24]. In addition, the larvae refine their control of swimming [22,25,26] and improve swimming performance [22,23]. These changes and improvements raise the following question: Do fish larvae produce swimming differently across early development and at different swimming speeds and accelerations?

To answer this question, we calculated bending moments for 113 swimming sequences of larvae aged between 3 and 12 days postfertilisation (dpf). The reconstructed bending moment patterns are qualitatively similar across development, speed, and acceleration. Rather than change the spatiotemporal distribution of bending moments, fish larvae control speed and acceleration with only the amplitude and duration of the bending moment patterns. This suggests that fish larvae retain the same relatively simple net actuation to produce swimming

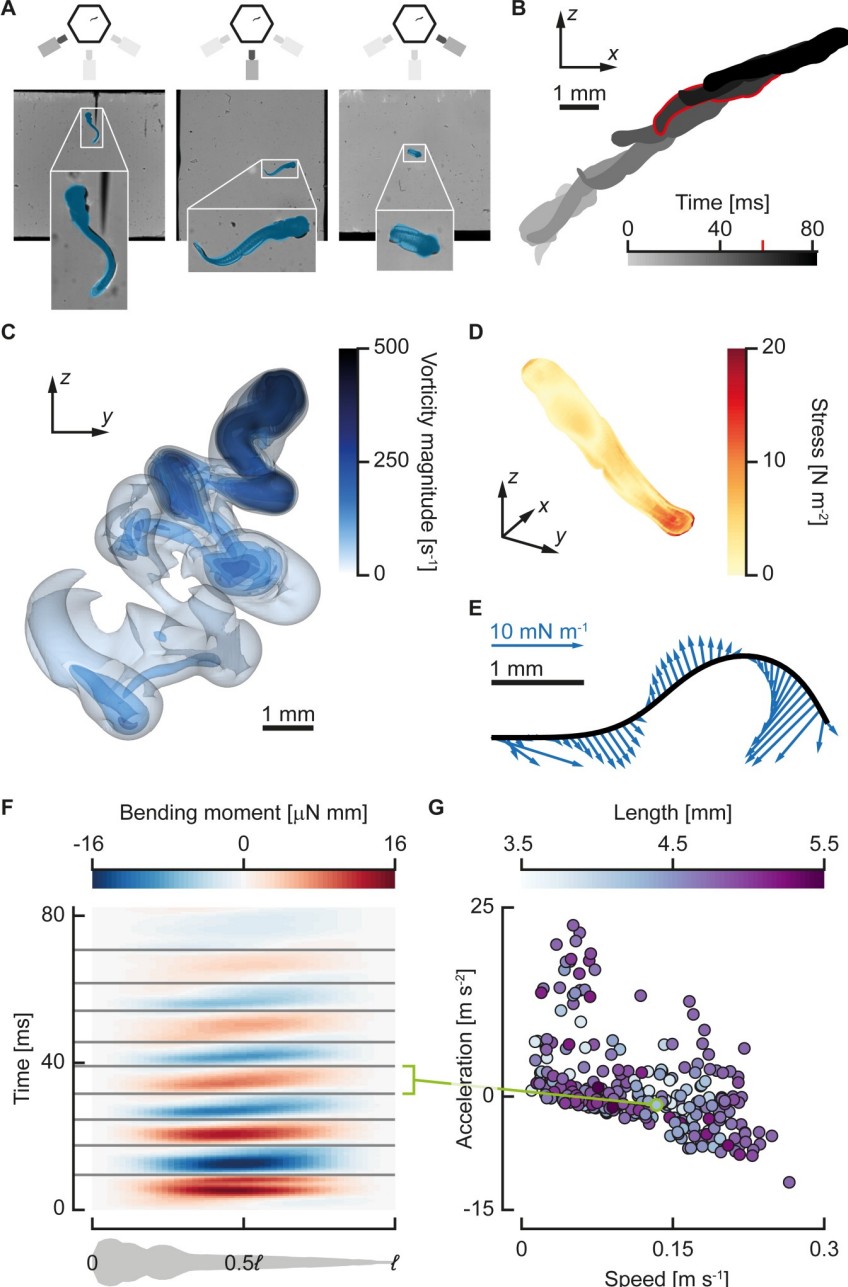

**Fig 1. Procedure for calculating bending moments.** (A) Larval zebrafish motion is reconstructed from synchronised three-camera high-speed video. Video frames (background) from the three-camera–setup overlaid with projections (blue) of the reconstructed model fish. The legend at the top indicates which camera produced the video frame. (B) Reconstructed three-dimensional motion from the video, projected onto the *x-z* plane; the highlighted time instant is shown in (A) and (C–E). (C) Transparent isosurfaces of vorticity for the same motion as (B) calculated with CFD. (D) Total fluid dynamic stress distribution on the skin, the magnitude of the sum of the pressure, and shear stress contributions calculated from the flow field from CFD. (E) Fluid dynamic force distribution transformed to the two-dimensional coordinate system attached to the deformation plane. This distribution is used as input to reconstruct internal moments and forces. (F) Reconstructed bending moment distributions (colour) along the fish (abscissa) and over time (ordinate). The grey lines separate the half phases in which the bending moment was divided. The green line links a single half phase to a data point in (G). (G) The mean speed (abscissa), mean acceleration (ordinate), and body length (colours) for individual half beats in the data set (*N* = 285). The green data point corresponds to the highlighted half beat in (F). Underlying data for panel G can be found in S1 Data. CFD, computational fluid dynamics.

across early development, despite the large neuromuscular restructuring throughout development and the complex physics that determines the resulting motion.

## Results

We have developed an inverse dynamics method for reconstructing bending moments of arbitrary three-dimensional swimming motion from multicamera high-speed video. The method comprises three main novel components: (1) three-dimensional motion tracking of freely swimming fish using machine vision techniques [21], (2) calculation of fluid forces acting on a swimming fish using computational fluid dynamics (CFD), and (3) bending moment reconstruction along a swimming fish using our large-amplitude beam model. An extended overview of the method is given in the Materials and methods section, and all (mathematical) details are provided in the S1 Text. Underlying data can be found in the accompanying data repository [27].

We highlighted the use and validity of our inverse dynamics method studying fish swimming using two approaches. First, we validated our methodology using a series of tests. Secondly, we used our method to investigate bending moment distributions of a large data set of swimming larval zebrafish.

### Testing and validation of our method

Validation of the methods and models used is described in detail in previous literature and the S1 Text. We validated the automated three-dimensional tracking method in a previous study [21]. In the S1 Text, section S5, we validate the moment reconstruction method with synthetically generated data. In S1 Text, section S6, we validated the CFD model used in the present method against a previously published, experimentally validated method [28–30], showing excellent qualitative and good quantitative agreement.

Next to these validations, we tested our large-amplitude beam model by comparing it with previous small-amplitude approaches [19,20,31]. In this comparison, we applied both beam models to the synthetically generated reference data that we used to validate our bending moment reconstruction method (S1 Text, section S5). These reference data consist of periodic swimming motions at three peak-to-peak tail-beat amplitudes (a small amplitude of $0.12\ell$, a medium amplitude of $0.20\ell$, and a large amplitude of $0.35\ell$; $\ell$ is body length).

The small-amplitude model showed error peaks of 7.6%, 16.4%, and 37.4% for the small-, medium-, and large-amplitude swimming motions, respectively (Figs 2 and 3). In contrast, our large-amplitude model showed error peaks of, respectively, 0.5%, 0.95%, and 5.1% at these same amplitudes (Figs 2 and 3). This showed that our large-amplitude model reconstructs bending moments with considerably lower errors at any amplitude but particularly so at larger amplitudes. Even at relatively small tail-beat amplitudes of $0.12\ell$, the present model shows errors of an order of magnitude smaller than the small-amplitude model. In addition to the differences in bending moment amplitude, the small-amplitude model also shows different dynamics; for example, the bending moment peaks are shifted in phase with respect to the reference, whereas the present model remains close to the reference in both phase and amplitude (Fig 2D).

The increasing error with amplitude is partly caused by the inaccurate small-amplitude motion model. This is illustrated in Fig 3C–3E, which shows the motion of the reference swimmer at its most extreme point represented by the small-amplitude model and the present model. Especially at larger amplitudes, it becomes necessary to consider the deformation in longitudinal direction to be able to reconstruct the motion accurately. The small-amplitude model changes centreline length considerably during the motion because of the lack of explicit

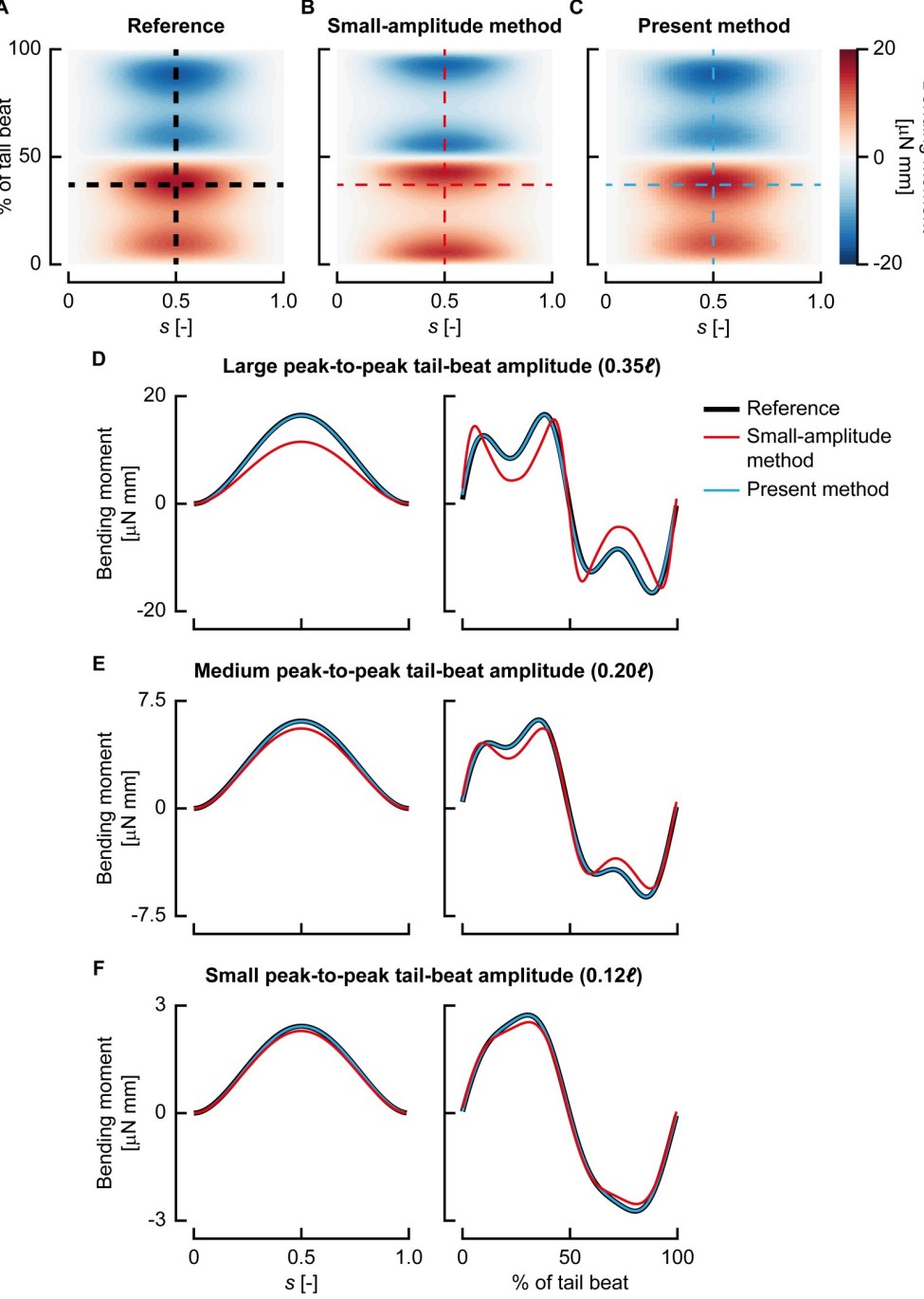

**Fig 2. Comparison of the present beam model with a small-amplitude model [17,19,20].** (A–C) Bending moment distribution (colours) along the length of the fish (*s*, abscissa) and over the tail-beat duration (ordinate) for the 'large-amplitude' reference solution (A), reconstructed with the small-amplitude model (B) and reconstructed with the present method (C). (D–F) Traces of the bending moment at a single time instant along the length of the fish (left panels; corresponding to the horizontal dashed line in [A–C]) and in the middle of the centreline over the tail-beat duration (right panels; corresponding to the vertical dashed line in [A–C]) for large-amplitude (D), medium-amplitude (E), and small-amplitude (F) motion.

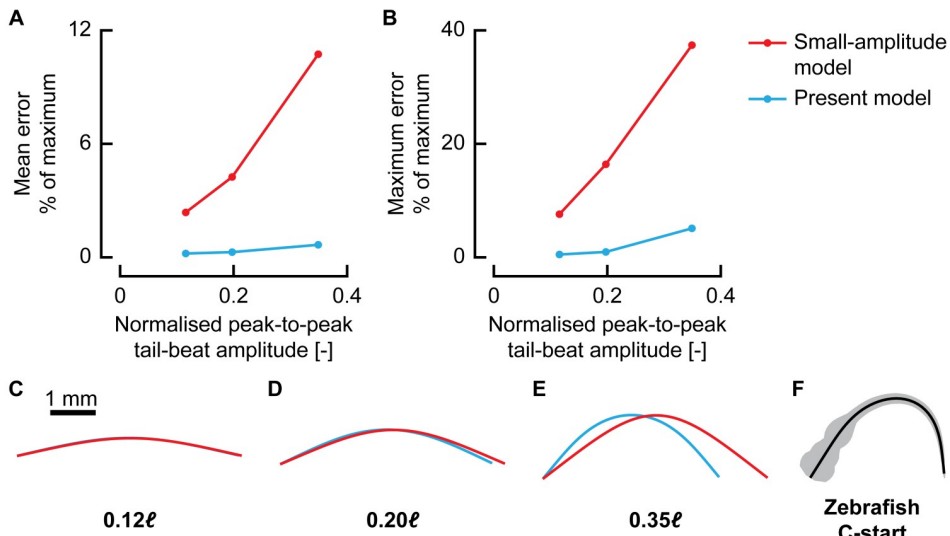

**Fig 3. Effect of motion amplitude on reconstruction accuracy.** (A–B) Average (A) and maximum (B) bending moment reconstruction error with respect to the maximum for the small-amplitude model (red) and the present method (blue) for increasing motion amplitudes. (C–E) Motion as represented by the small-amplitude model (red) and the present method (blue) for increasing motion amplitude. Note that because the small-amplitude model only uses a lateral displacement, the centreline changes length throughout the motion. (F) Example body shape of a 3-days-postfertilisation zebrafish larva performing a C-start. Underlying data for panels A and B can be found in S1 Data.

modelling of deformation in *x* direction. In addition to this motion simplification, the small-amplitude model also assumes that the deformation angles are close to zero (and hence sin $\theta \approx \tan \theta \approx \theta$; cos $\theta \approx 1$); this is not the case for the zebrafish of this study, as the angles are often well outside the range of validity of this assumption (e.g., Fig 3F).

The largest amplitude in our comparison is still considerably smaller than the tail-beat amplitudes exhibited by the zebrafish larvae (Fig 3F); thus, for these larvae, the small-amplitude model would produce even larger reconstruction errors of bending moments. A similar problem occurs for adult fish. Although they tend to swim at lower relative motion amplitudes than those of larval fish, during (near-)periodic swimming, many are still in the range of 'medium' tail-beat amplitudes [32]. Hence, even for adult fish swimming with moderate tail-beat amplitudes, dropping the small-amplitude assumption results in much-reduced errors.

In addition to the small-amplitude assumption, the model used in previous studies [17,19,20] assumes periodic motion. There is hardly close-to-periodic motion in our experimental data, necessitating a model that drops this assumption. In general, aperiodicity is common in fish swimming; in many cases, the aperiodic motions, e.g., fast starts, turns, and forward acceleration, are most interesting to analyse bending moments for.

## Demonstration of the method: Bending moments in larval zebrafish

To illustrate how the method can be applied, we used it to investigate bending moment distributions of a large data set of swimming larval zebrafish. The results of this example analysis are presented in this section.

**Example analysis of the bending moments of a larval swimming bout.** Fig 1 shows an example analysis of a single swimming bout of a larval zebrafish. We analysed the body motion from the footage of three synchronised high-speed video cameras with an automated method [21] (Fig 1A and 1B). We then used these data to compute the flow field around the fish with a

CFD method, which is visualised at one moment during the event with vorticity isocontours (Fig 1C). The flow field was postprocessed to calculate the fluid dynamic stress on the body surface (Fig 1D, same instant as Fig 1C) and the fluid dynamic force distribution along the fish axis (Fig 1E, same instant as Fig 1C). Finally, we combined all these data to compute the bending moments along the body during the swimming bout (Fig 1F).

**Overview of an individual swimming sequence.** We performed phase averaging on a periodic section of a swimming sequence of a 3-dpf zebrafish larva to illustrate how bending moments and bending powers vary along the body during swimming. We selected four half tail beats (Fig 4A) based on the periodicity of the body curvature. We averaged body curvature, bending moment, kinetic power, and fluid power over these half beats.

Body curvature (Fig 4B and 4C) shows a travelling wave pattern behind the stiff head with one positive and one negative peak per cycle. The highest curvatures are reached near the tail, at around 80% of the body length ($\ell$), where body width is relatively small. Curvature waves originate from around $0.25\ell$, which is close to where the most anterior axial muscles are located. They then travel at approximately constant speed ($3.3\ell$ per tail beat) posteriorly, growing in amplitude until close to the tail, and finally dropping to zero amplitude at the tail tip.

Bending moments (Fig 4D) show a positive and a negative peak during swimming, corresponding to the direction of the tail beat, but preceding it in phase along most of the body. The peak amplitude occurs around $0.4\ell$, corresponding to the area with the highest muscle cross section (Fig 4E). Bending moments in the head and tail regions are low because of the free-end boundary conditions (in which the bending moment must be zero); the absence of muscle; and in the tail region, the limited cross-sectional area. Like curvature, the bending moment also shows a travelling wave pattern, but its wave speed is more than twice as high as the curvature wave speed ($7.1\ell$ per tail beat).

The power used by the body to move the fluid (Fig 4F) shows a large peak close to the tip of the tail. The motion amplitude is large here (Fig 4A and 4B), as well as the lateral velocities; therefore, fluid forces are large. Because power is the product of velocity and force, most power is expected to be transferred to the fluid here. The kinetic power, defined as the time rate of change in kinetic energy of the body, is smaller in magnitude than the fluid power (Fig 4G). The head shows considerable variation in kinetic energy over a tail-beat cycle owing to its relatively large mass and side-to-side motion. There is a dip in kinetic energy fluctuations in the anterior region of the yolk sac. In the remainder of the body, the kinetic power shows a travelling wave pattern caused by the travelling wave character of the body motion and, hence, its speed. The resultant power (Fig 4H), defined as the sum of the fluid and kinetic power, is dominated by the fluid power.

**Swimming effort and vigour.** We reconstructed three-dimensional kinematics from 113 video sequences of fast-start responses followed by swimming, calculated flow fields throughout the sequence with CFD, and fitted distributions of internal forces and moments. These swimming sequences hardly contain periodic swimming. To analyse the data despite their aperiodicity, we subdivided them into half beats based on zero crossings of the bending moment in the midpoint along the centreline (Fig 1F and 1G). For each of these 285 half beats, we calculated the period length, mean speed, mean acceleration of the centre of mass to the next half beat, and peak (95th percentile) bending moment.

To reduce the number of parameters for the analysis, we identified combinations of parameters with high explanatory capacity—one parameter related to the 'input' delivered by the fish and one related to the 'output' of the swimming motion.

To control swimming, the fish has two main input parameters to change the bending moment (see section below): the duration of the half beat ($t_{\text{half}}$) and the bending moment amplitude (i.e., the peak bending moment $M_{\text{peak}}$). We define the 'swimming effort' as

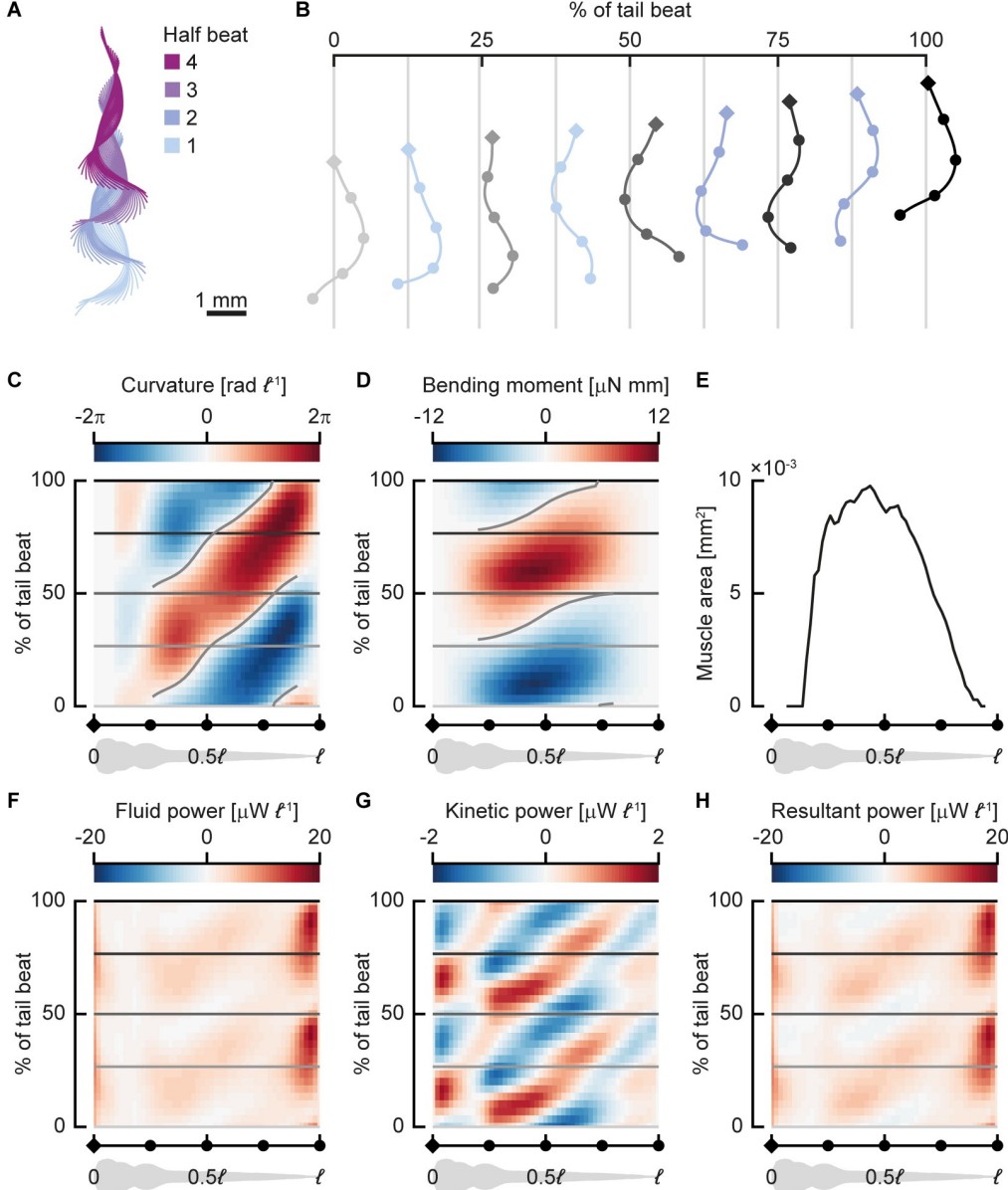

**Fig 4. Near-periodic sequence of a 3-days-postfertilisation zebrafish larva.** The larva swam at $31\ \ell\,\mathrm{s}^{-1}$ with a tail-beat frequency of 69 Hz. (A) Centreline motion throughout the sequence. The colours indicate half phases. The coordinates were transformed to a best-fit plane through all points along the centreline throughout the motion. (B) Motion during a single full tail beat (half beat 1 and 2) of the swimming sequence; the grey centrelines (from light to dark grey) correspond to the time points shown with horizontal lines in (C, D) and (F–H). The diamond (head) and dots on the centrelines correspond to points on the abscissa of (C–H). (C, D, F–H) Heat maps of distributions (colours) along the fish (abscissa) and over the phase over the tail beat (ordinate); all quantities are averaged over separate half beats; 'negative' half beats are mirrored for the curvature and bending moment. (C) Body curvature normalised by body length. (D) Bending moment. (E) Transverse muscle area distribution along the fish. (F) Fluid power per unit body length (power exerted by the fish to move the fluid). (G) Kinetic power (rate of change in kinetic energy) per unit body length; note the difference in scale with panel F. (H) Resultant power per unit length; the sum of the fluid and kinetic power. Underlying data for panel E can be found in S1 Data.

$E = M_{\mathrm{peak}}\ t_{\mathrm{half}}^{-1}$—higher bending moments and shorter periods increase $E$. We fitted a generalised linear model (gamma distribution, log link function) with MATLAB (fitglm, R2018b, The Mathworks) and the Statistics and Machine Learning Toolbox (R2018b, The Mathworks).

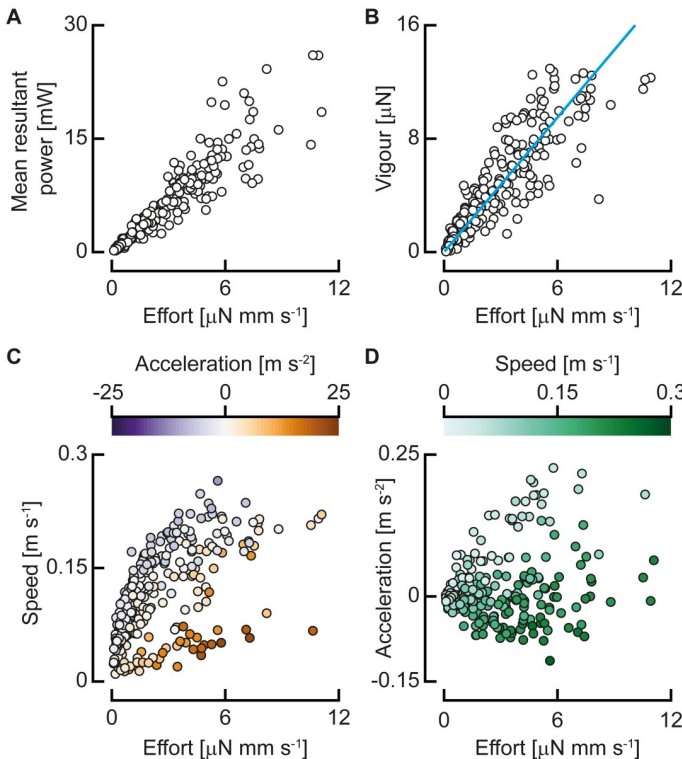

**Fig 5. Swimming effort and vigour for all analysed half tail beats.** (A) The mean resultant power over a half beat as a function of swimming effort ($N = 398$). (B) Swimming vigour as a function of effort, and the optimal linear fit ($N = 285$). (C) Mean speed over a half beat as a function of swimming effort, with colour indicating the mean acceleration ($N = 285$). (D) Mean acceleration over a half beat as a function of swimming effort, with colour indicating the speed ($N = 285$). Underlying data for panels A–D can be found in S1 Data.

This showed that the swimming effort correlates significantly with the mean resultant power (Fig 5A; $P < 0.0001$), with an exponent of 1.06—close to linear.

The most important output parameters for swimming motion are the speed and acceleration, so we define a summarising parameter that includes both. We expect the net propulsive force to scale with the mass of the fish, its acceleration, and its squared speed (from the dynamic pressure). We can give an indication of the strength of the output of a swimming motion by looking at the estimated amount of required propulsive force for the combination of speed and acceleration. Based on this, we define an output parameter swimming vigour as $V = m(cv^2 + a)$, where $m$ is body mass, $v$ is swimming speed, and $a$ is mean acceleration (i.e., change of speed to the next half beat per unit of time). The coefficient $c$ is calculated with an optimisation algorithm that minimised the sum of squared errors of a linear fit of vigour to effort with total least squares. The fitted value of 517.7 m$^{-1}$ results in a clear trend of vigour as a function of effort (Fig 5B; generalised linear model fit with gamma distribution and log link function: $P < 0.0001$), collapsing the broad clouds of speed and acceleration (Fig 5C and 5D).

**Bending moment distributions are similar across swimming vigour and development.** To assess how bending moment patterns differ across vigour and development (indicated by body length [24]), we compared bending moment patterns normalised by their amplitude and duration. This allowed us to compare bending moment patterns of half beats performed with different swimming effort and assess whether a common spatiotemporal pattern exists that is independent of the duration and amplitude. We normalised the bending moment distribution

of each half beat by dividing by the peak bending moment. We then calculated the mean and standard deviation (Fig 6A and 6B) of the normalised distributions of all half beats. The standard deviation (Fig 6B) is relatively small, locally peaking at 0.24, which is caused primarily by variation in the peak phase (Fig 6E and 6F).

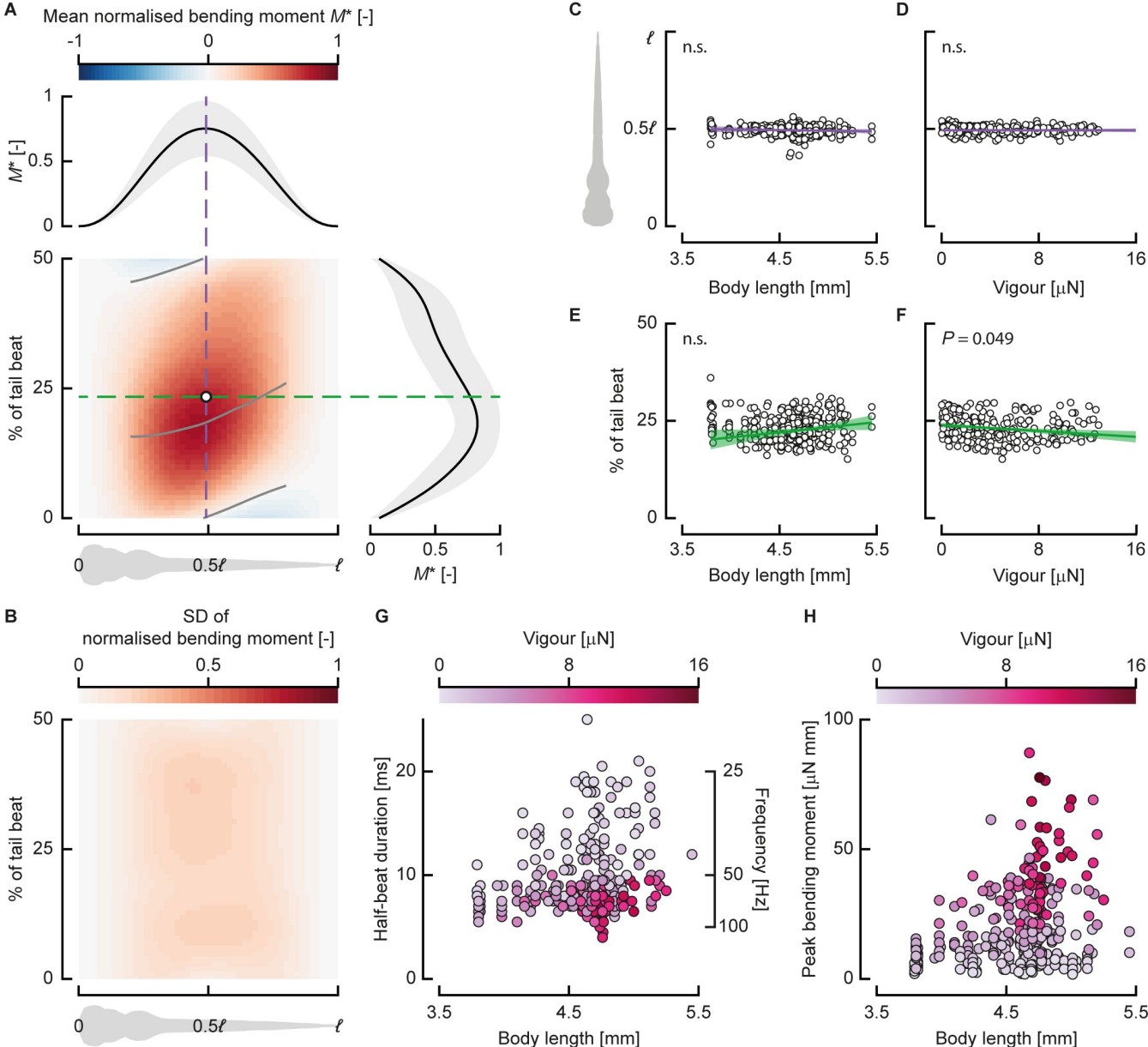

**Fig 6. Bending moment patterns are similar across swimming vigour and development.** (A) Normalised bending moment pattern along the fish (abscissa) and over normalised time (ordinate) averaged over all half beats ($N = 398$). The dashed lines indicate slices through the pattern in time (green) and location (purple) of the centre of volume of the distribution, shown respectively at the top and right of the heat map, along with their standard deviation. The grey lines over the heat map show the zero and maximum contour line for the middle 60% along the body. (B) The SD of the normalised bending moment along the fish (abscissa) and over normalised time (ordinate) ($N = 398$). (C, D) The location along the body of the centre of volume of the bending moment as a function of body length ([C] $N = 398$) and swimming vigour ([D] $N = 285$). (E, F) Normalised time of the centre of volume of the bending moment as a function of body length ([E] $N = 398$) and swimming vigour ([F] $N = 285$). (G) Half-beat duration as a function of length (i.e., developmental stage), with colour indicating swimming vigour ($N = 285$). (H) Peak bending moment as a function of length, with colour indicating swimming vigour ($N = 285$). Underlying data for panels C–H can be found in S1 Data. n.s., not significant; SD, standard deviation.

To further assess the spatiotemporal similarity of the patterns, we calculated the mean absolute difference of each point in the bending moment distribution to the corresponding point in the mean distribution—effectively the overall difference of each individual pattern from the mean over all patterns. The mean of these differences over all half beats is 0.091 ± 0.028—the differences are relatively low and of similar magnitude across half beats. Thus, the patterns look similar across different developmental stages and swimming vigour. Note that the peak value is smaller than 1 (Fig 6A) because the peak location shows variation in both location along the body and phase (Fig 6C–6F).

To analyse how the bending moment distributions change with vigour and body length (i.e., developmental stage), we calculate the centre of volume of the bending moment distributions. We computed this as the average location along the body and time in the half beat, weighted by the bending moment at each point along the body and time instant; this is equivalent to calculating the centre of mass along two axes. The centre of volume of the individual bending moment patterns (Fig 6C–6F) lies around $0.5\ell$ along the body length and 25% of the tail beat (i.e., 50% of the half beat). The location along the body varies little across body length and swimming vigour. The phase (i.e., time relative to the tail-beat duration) shows more variation over body length and vigour but shows no clear pattern. We fitted linear models with MATLAB (fitlm, R2018b, The Mathworks) with the centre of volume location along the body and in phase as response variable and body length and vigour as predictors. The slopes for the centre of volume position along the body are not significantly different from 0 for length ($P = 0.071$), vigour ($P = 0.78$), or their interaction ($P = 0.78$). The slopes for the phase of the centre of volume are not significantly different from zero for length ($P = 0.32$) and the interaction between length and vigour ($P = 0.065$) but are marginally significant for vigour ($P = 0.049$).

Although the spatiotemporal distributions (i.e., wave length, amplitude distribution, etc.) of the bending moments are similar across lengths (i.e., developmental stage), the duration and amplitude vary (Fig 6G and 6H). As the fish develop, the range of half-period durations increases (Fig 6G)—young larvae use mostly short durations, whereas larger larvae use a broad range of durations. The maximum peak bending moment increases substantially over development (Fig 6H). Older fish can generate higher peak bending moments and can reach higher swimming vigour values, but they do not always do so.

**Control parameters of swimming vigour.** Because the bending moment patterns are similar across swimming styles and developmental stage, the parameters left for controlling swimming vigour are the amplitude of the bending moment and the duration of the tail beat. All experimental points lie on a broad cloud around a curve through the effort landscape, which is a function of peak bending moment and half beat duration (Fig 7A–7C). In general, high peak bending moments are only produced for tail beats of short duration. As the duration decreases (i.e., frequency increases), the bending moment amplitude decreases. Higher efforts generally lead to higher speeds (Fig 7B), unless the larva is accelerating strongly. Strong accelerations are mostly found with slow-swimming larvae that use short half-beat durations and high peak bending moments (Fig 7A). For high-effort tail beats, the larvae are generally either swimming fast or accelerating: high-effort tail beats at low swimming speeds show high accelerations, whereas high-effort tail beats at low accelerations show high speeds. As a result of the curve fit, swimming vigour increases with increasing effort (Fig 7C, see also Fig 5B).

Effort can be increased either by increasing the peak bending moment or by decreasing the half-beat duration, or a combination of both. To assess how larvae vary these parameters, we divided the half beats into three bins of effort, each with the same number of points (Fig 7D and 7E). For each of these bins, we show the frequency distribution of the peak bending moment (Fig 7D) and the half-beat duration (Fig 7E). Low-effort half beats are usually done

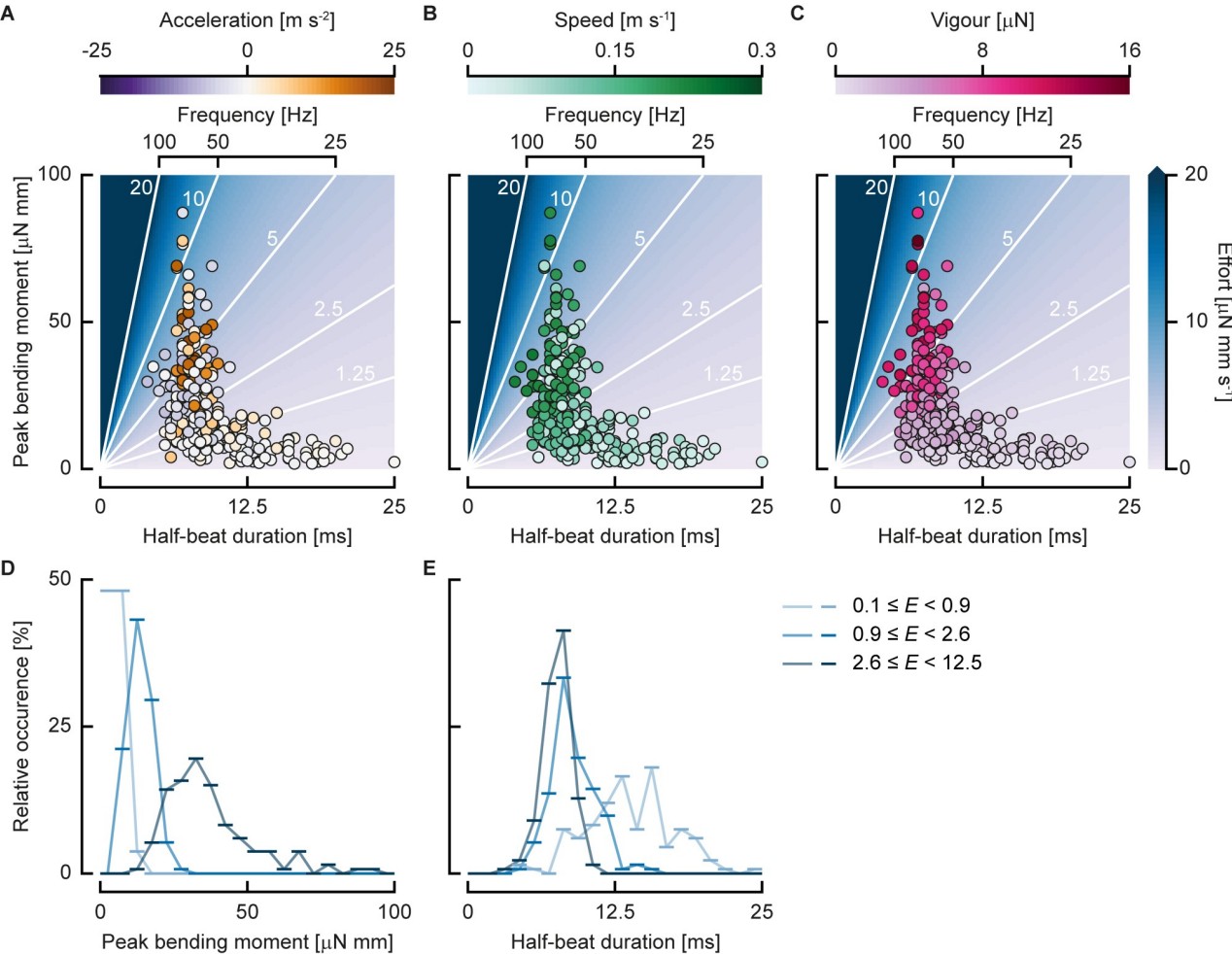

**Fig 7. Swimming control parameters.** (A, B, C) Individual half beats in the duration–peak bending moment landscape. The coloured background with white contour lines shows the swimming effort (*N* = 285). (A) Dot colour indicates acceleration. (B) Dot colour indicates speed. (C) Dot colour indicates swimming vigour. (D, E) Relative occurrence of half beats in bins (line markers) of peak bending moment (D) and half-beat duration (E) divided in 3 bins of swimming effort (colours), each containing the same number of half beats. Underlying data for panels A–E can be found in S1 Data.

with low peak bending moment and relatively long duration with a large spread. In contrast, medium and high-effort half beats tend to occur at similar half-beat durations. The main difference between the middle- and highest-effort bin is in the peak bending moment.

## Discussion

In this study, we analysed the actuation used during swimming of developing zebrafish larvae by calculating bending moment patterns with inverse dynamics. We found that zebrafish larvae use similar bending moment patterns across development. They adjust their swimming vigour, a combination of speed and acceleration, by changing the peak bending moment and tail-beat duration. At higher speeds and accelerations, the larvae produce the required fluid dynamic forces by increasing bending moment amplitude and/or decreasing tail-beat duration.

Previous inverse dynamics approaches for the internal mechanics of swimming used simplified models for both the fluid and the structure. The structure was modelled with linear bending theory, assuming small deformations of the centreline [17,19]. The effects of a large-

amplitude correction to these were expected to be small for periodically swimming adult fish [33]. However, fish larvae, as well as turning and fast-starting adult fish, beat their tails often at >90˚ with the head [23,34,35], violating the small-amplitude assumption.

We compared the previously used small-amplitude beam model [17,19,20] with the new large-amplitude beam model. This showed that our large-amplitude model reconstructs bending moments with considerably lower errors at any amplitude but particularly so at larger amplitudes. The motion amplitude can be summarised with the peak-to-peak tail-beat amplitudes normalised by fish length $\ell$, which is usually in the range of 0.1–0.3 for adult fish [32]; for larval zebrafish, it reaches values above 0.5 [23]. At a moderate peak-to-peak tail-beat amplitude of $0.2\ell$, the small-amplitude model showed error peaks of 16.4%, whereas at $0.35\ell$, this had increased to 37.4%. In contrast, our large-amplitude model showed error peaks of, respectively, 0.95% and 5.1% at these same amplitudes. This indicates that the small-amplitude model is inadequate for analysing bending moment distributions of large-amplitude swimming, which occurs in larval fish and many adult fish. Our beam model does ignore the effect of shear deformation that is expected to occur close to the medial plane [36], but we expect it to be of small influence on the bending moments because of its proximity to the axis and hence small moment arm.

In addition to the large-amplitude beam model, we used three-dimensional CFD to calculate fluid dynamic forces, dropping previous assumptions of inviscid flow [17–19,37] and the necessity to model the boundary layer separately [33]. The assumption of inviscid flow does not hold for fish swimming in the intermediate regime [3,28]—full solution of the Navier–Stokes equations is necessary to obtain sufficiently accurate fluid dynamic force distributions. The intermediate Reynolds number of the zebrafish larvae (45–1,500 in our experimental data) allows us to solve the Navier–Stokes equations accurately, without requiring turbulence modelling [28]. The main limitation on the accuracy of the computed flow fields is the input motion that we reconstructed from high-speed video. However, we expect the resulting error to be limited because the overall motion of the fish is reconstructed accurately, despite local errors in curvature (e.g., near the tail) [21].

In our analysis of swimming sequences across development, we do not assume periodicity. Periodic motion is a common assumption in the analysis of fish swimming [23,31]. For zebrafish, (near-)cyclic swimming occurs most often after a fast start, generally only for a few tail beats, and is rarely spontaneous [22,38]. To analyse aperiodic motion, we subdivided each swimming sequence in tail beats based on zero crossings of the bending moment in the middle of the body. During aperiodic swimming, the mean speed varies between successive tail beats. For this reason, we define a parameter, swimming vigour, that combines the effects of acceleration and swimming speed as $V = m(cv^2+a)$. This approach for analysing aperiodic swimming could be of general use in swimming research. The subdivision in half beats can also be done with quantities other than the bending moment—for example, body curvature. This enables similar analyses of aperiodic swimming from pure kinematics without inverse dynamics.

We define the swimming effort exerted by the fish based on the amplitude of the bending moment and the duration of the tail beat as $E = M_{\text{peak}} t_{\text{half}}^{-1}$. This quantity correlates with resultant power, showing that it is indeed an indicator of amount of effort exerted by the larva (Fig 5A). The speed and acceleration fall on broad clouds as a function of effort (Fig 5C and D) because the required power depends on their combination rather than their individual values. The effort–speed landscape (Fig 5C) shows a two-pronged distribution, with one branch showing high effort but low speed and the other, broader branch showing increased effort with speed. This distribution is mainly explained by the acceleration, showing high values in the lower branch—fish only accelerate strongly from low speeds and use high effort to do so. This

is reflected in the effort–acceleration landscape (Fig 5D), with low (including negative) acceleration being found at high speeds, and vice versa.

When speed and acceleration are combined into the swimming vigour, these clouds collapse closely to a trend line (Fig 5B). Remaining variation in this trend may be partly caused by turning behaviour and contributions of the pectoral fins. The coefficient $c$ determines the relative contribution of the speed and acceleration to the swimming vigour, giving an indication of their relative cost.

To gain more insight into these relative contributions, we assume that the contribution of the velocity to the vigour is equal to $c_d \frac{1}{2} \rho S v^2$, where $c_d$ is the drag coefficient, $\rho$ is the fluid density, and $S$ is the wetted area of the fish [30]. Furthermore, we assume that the contribution of acceleration is given by $m(1+c_m)a$, where $c_m$ is the added mass coefficient [39,40]. The drag coefficient was calculated to be 0.26 in a previous CFD study on larval zebrafish [30]. Given that the vigour coefficient $c$ is the ratio between the velocity contribution and the acceleration contribution, we can derive the added mass coefficient to be 3.26. This indicates that the fish experiences a fluid acceleration reaction upon accelerating that is 3.26 times larger than its own mass. For a flat plate accelerating perpendicular to the flow, the theoretical added mass coefficient is 1 [39], and for eels swimming at a Reynolds number of 16,000–120,000, the added mass coefficient was estimated to be 2.8 ± 0.8 [40]. Although our larvae fish swim at a much lower Reynolds number than the eel, their added mass coefficient is thus similar.

Most of the resultant power produced by the fish during near-periodic swimming is used to increase the energy in the fluid rather than the kinetic energy of the body (Fig 4H). The energy spent on the water is likely lost on lateral velocity: larvae swim at high Strouhal number, which is associated with large tail-beat amplitudes and relatively high energy consumption [23,41]. Most of this fluid power is produced at the tail, where the largest fluid dynamic forces are produced [30], even though no muscles are present here. This suggests a transfer mechanism by passive tissues from the muscles to the tail [42–44]. The ratio of kinetic power of the body and resultant power might be higher during fast starts and hence deserves further research.

The bending moment does not correspond directly to muscle action because it also includes the effects of passive structures inside the body [17]. We do not know the contribution of the muscles to the total bending moment, nor do we know the specific distribution of stresses inside the body. Cheng and colleagues [20] modelled the elastic and viscoelastic properties of the passive tissue and thus estimated the contribution of the muscle bending moment. The amplitude of the muscle bending moment was found to be higher than the overall bending moment, whereas the wave speed was found to be lower. However, the overall dynamics look reasonably similar. If we assume similar distributions of passive tissues inside the fish across the considered developmental stages [24], similar total bending moment patterns will require a similar muscle contribution. Furthermore, the difference in amplitude between similar bending moment patterns must originate from the muscle moment because it is the only net source of power in the system—the work done by the fluid and passive tissues indirectly comes from the muscles.

We found that the bending moments follow a similar pattern across development and swimming vigour (i.e., speed and acceleration). The only significant coefficient in the linear models is the phase of the centre of volume of the bending moment patterns as a function of swimming vigour (Fig 6F), but the effect is limited. More vigorously swimming fish generate the peak bending moment slightly earlier in the half beat.

The mean bending moment pattern looks qualitatively similar to earlier calculations done for adult fish [19]. It is a single-peaked distribution, with the maximum around the bulk of the muscle (Fig 4E and Fig 6A) and a fast-travelling wave character. Muscle electromyograms

(EMGs) during visually induced fictive swimming done on paralysed zebrafish looked similar to activation patterns in adult fish [13]. This suggests that this simple pattern of bending moments is common to fish across species and developmental stage. Even though fish larvae swim in the intermediate regime [3], and adult fish often swim in the inertial regime [45], the differences in fluid dynamics do not seem to require fundamentally different bending moment patterns.

For accelerating motions, swimming kinematics have been observed to be similar and hydrodynamically optimal across a wide range of body shapes and sizes [46]. In bluegill sun-fish, muscle activation patterns have been measured during acceleration, showing that they effectively stiffen their bodies by coactivating muscles on both sides of the body [47]. This raises an interesting question: Is this activation pattern conserved across fish species to effectively produce acceleration? Our method could help answer this question by making it possible to reconstruct internal mechanics from only kinematics, without requiring invasive EMG.

Because the bending moments look similar along the body and over the phase for each half beat, the two parameters left for the larvae to adjust for each half beat are the amplitude of the bending moment and its duration (Fig 7D and 7E). Young larvae use a relatively narrow range of amplitudes (Fig 6H) and half-beat durations (Fig 6G), which broadens as the fish develop. Older larvae are able to generate higher peak bending moments, which is likely correlated to development of their muscle system [48]. Furthermore, older larvae use a broader range of tail-beat durations than young larvae, suggesting that older larvae have more freedom to control their swimming vigour. The relative contributions of the peak bending moment and half-beat duration depend on the magnitude of the effort. At low efforts, a wide range of half-beat durations is used, but when the effort is increased, this parameter saturates (Fig 7E) as the larvae approach their maximum muscle contraction frequency of 90–100 Hz [22,23]. For further increase of the effort, the peak bending moment must increase (Fig 7D).

Swimming kinematics emerge from simple bending moment patterns. These patterns presumably stem from simple muscle activation input—their quantification is an interesting avenue for future research. The arrangement and properties of the muscles, passive tissues, and propulsive surface cause simple inputs to translate into complex kinematics and flow fields. This has profound consequences for larvae that need to swim to survive [49]. Straight from the egg, larval fish can produce swimming behaviour to escape threats despite relatively limited neural processing capacity. This concept of designing passive systems to allow complex systems to be actuated simply is of broad interest in engineering and biology [50–52]. Because bony fish larvae are similar in morphology across many species [53], we expect that these results are relevant for bony fish in general. The simple actuation solution may have been instrumental in the evolution and adaptive radiation of bony fish, with more than 30,000 extant species.

We demonstrated the novel methodology on the specific example of larval zebrafish. There are numerous other potential applications for this method in the field of fish swimming. For example, an extensive analysis of turning fish could clarify how they produce this turning motion. In a recent study of fast-starting larval fish [54], it was found that fish larvae can independently control their escape speed and turning angle. The here-developed methodology could help to elucidate the required underlying actuation patterns. Another interesting application would be to analyse a large data set of adult fish swimming at different speeds and accelerations. This would allow assessment of whether adult fish also use similar bending moment patterns across speeds, whether these patterns are similar to larval fish, and how these patterns are affected by body morphology (i.e., distributions of muscle fibres and passive tissue components). The combination of analysing three-dimensional kinematics and fluid dynamics and use of a large-amplitude deformation model not only opens the doors to a broad range of

studies into the fluid–structure interaction of fish swimming but also to other undulatory swimmers, such as aquatic snakes and worms.

## Materials and methods

Here, we summarise the main steps of our methodological approach. An in-depth mathematical treatment of the methods is given in S1 Text. Source codes of the computational methods can be found in the accompanying data repository [27].

### Ethics statement

The experiment complies to the Dutch Act on Animal Experiments, which complies with European Directive 86/609/EEC, and was approved by the animal welfare authority (DEC Wageningen University, the Netherlands). The institutional licence number to conduct animal experiments is 10400. The protocol number of the approved experiment is 2013087.b, projectcode LarvalSwim, experimental code 2013096.

### Reconstructing three-dimensional motion from multicamera high-speed video

We made high-speed video recordings of fast starts of three separate batches of 50 zebrafish larvae from 3–12 dpf. The camera setup was identical to the setup described in Voesenek and colleagues [21], with three synchronised high-speed video cameras recording free-swimming larvae at 2,000 frames per second. To reconstruct the swimming kinematics from the recorded high-speed video, we used automated three-dimensional tracking software developed in-house [21] in MATLAB (R2013a, The Mathworks). For every time point in a multicamera video sequence, the software calculated the best fit for the larva's three-dimensional position, orientation, and body curvature to the video frames. These parameters were then used to calculate the position of the larva's central axis and the motion of its outer surface (Fig 1A and 1B).

### Subdividing motion

We calculated phase-averaged quantities for an individual swimming sequence to look in detail at the generated bending moments and powers. We determined whether a (subset of a) sequence is periodic with a similar approach to Van Leeuwen and colleagues [23]. For every possible subset of a swimming sequence, we calculated the sum of absolute difference with a time-shifted version of the curvature, similar to an autocorrelation. We then calculated extrema in this function—if extrema are detected, their maximum value determines the 'periodicity' of the sequence. We then selected the longest possible subsequence that has a periodicity value higher than a threshold of 35—this is a swimming sequence for a 3-dpf fish. We divided this sequence into half phases based on peaks in the body angle [21,23]. The curvature, bending moment, fluid power, kinetic power, and resultant power were then phase averaged based on these subdivisions.

Most of the swimming of larval zebrafish is aperiodic, and long bouts of tail beats at the same frequency and amplitude are rare. Therefore, the common approach of analysing fish swimming as a periodic phenomenon is therefore not appropriate. As an alternative, we divide each swimming bout in half tail beats, and for each, we can determine a mean swimming speed and acceleration. We mirror all half beats toward the left side of the fish such that all extracted half beats were in the same direction. This approach allows us to systematically analyse aperiodic motion, opening up a much larger fraction of the data set for investigation.

We used the bending moment patterns to divide the swimming motion into half tail beats —we defined the start of each half tail beat as the moment at which the bending moment at $0.5\ell$ crossed the zero line. Since some of these zero crossings are related to noise, we evaluated every possible permutation of zero crossings per sequence on several criteria with a custom MATLAB (R2018b, The Mathworks) programme. We eliminated zero crossings with neighbouring sections with an amplitude of less than 5% of the peak half-beat amplitude in the sequence because they are most probably noise. We required more than three zero crossings to have at least two half beats to be able to calculate a mean acceleration. Extreme values in each half beat should alternate direction to eliminate noisy zero crossings: the larva beats its tail left and right; therefore, bending moment must alternate. Finally, we eliminated half beats with a duration shorter than 2.5 ms (equivalent to 200-Hz tail-beat frequency)—the maximum tail-beat frequency observed for zebrafish larvae is 95 Hz [23]. From all permutations that met the criteria, we selected the permutation with the smallest standard deviation in half-period length across the sequence. This left the longest possible, least noisy sequence of half beats for every swimming bout.

Out of 113 swimming sequences, we selected 398 half beats with this procedure. For each of these half beats, we calculated the duration, mean speed, and peak bending moment. We determined the mean acceleration by calculating the difference in mean speed (i.e., velocity magnitude) between the following and current half beat and dividing by the time difference between the tail-beat midpoints. Because we could not calculate mean acceleration for the last half beat in each sequence, 285 half beats remained for which we computed all quantities.

## Calculating fluid force distributions

The equations of motion include an external force distribution, which is produced by the water on the skin. Because this distribution is exceedingly difficult to measure directly and noninvasively, we modelled the fluid dynamics numerically. Previous approaches used inviscid fluid models to calculate fluid force distributions [17,19,20]. However, at the intermediate Reynolds number regime of the larval fish, these models are inaccurate because a considerable fraction of the forces are generated through viscous effects [30]. Instead, we perform CFD to calculate solutions of the full Navier–Stokes equations. This results in an accurate representation of the fluid force distributions because all flow scales can be represented numerically [28]. We used two independent CFD methods to calculate fluid dynamic forces.

We performed CFD using a Navier–Stokes solver based on overset meshes [28–30] coupled to a body dynamics solver to simulate free swimming. Simulations were performed with swimming kinematics based on a travelling wave with a known curvature amplitude at a frequency of 50 Hz. The same motion was used in a second independent Navier–Stokes solver based on the immersed boundary method, the open-source code IBAMR [55]. We used an experimentally validated method to validate the second method to assess its accuracy when calculating internal forces and moments.

The Navier–Stokes equations were solved on a rectangular domain, with extents determined by the bounding box around the complete motion with an additional margin of two fish lengths. The immersed boundary solver used an adaptive mesh refinement approach in which the computational mesh can be locally refined depending on the flow conditions. In our case, the mesh consisted of four levels of refinement. Each level was a simple rectangular Cartesian mesh with four times the number of subdivisions in all dimensions compared with the coarser level. The choice of mesh refinement level depended on the local value of the vorticity; we chose thresholds of absolute vorticity of $1\ \text{s}^{-1}$, $25\ \text{s}^{-1}$, and $250\ \text{s}^{-1}$ to switch to the second, third, and fourth refinement levels, respectively. We used a fixed time step of 0.5 μs (see section

6 in S1 Text), in which the Courant–Friedrichs–Lewy (CFL) number is always much smaller than 1 (i.e., the fluid moves much less than a grid step in a single time step). We saved the fluid solution every 0.25 ms, at which we reconstructed the internal forces and moments.

The surface of the fish was described as a cloud of Lagrangian points moving over the Eulerian fluid solution mesh. The motion of these points was prescribed based on quintic spline interpolation [56] of the tracked kinematics with a custom-developed add-on to IBAMR written in C++. This add-on reads the three-dimensional tracked data and interpolates it to the solver time steps. It then uses these data to calculate the position and velocity of a point cloud that was generated based on the three-dimensional surface model. We use a hollow point cloud describing the surface to prevent the existence of a nondivergence-free velocity field inside the fish that could disturb the external flow field. The density of this point cloud was chosen such that the mean distance between the points is 75% of the density of the smallest mesh level. This ensured that each mesh cell inside the fish body will have at least one body surface point in it and not much more.

Because we use a hollow point cloud that allows for flow inside the fish, we cannot directly use the immersed boundary forces on the Lagrangian points to calculate the force distribution. Instead, we postprocess the resulting three-dimensional velocity and pressure fields (Fig 1C) to extract the fluid force distribution on the skin of the fish with a custom Python 3 programme. In this programme, we interpolated the pressure and velocity gradients to a triangulated surface slightly offset from the fish skin [57]. We then integrated these values into contributions to the pressure force and the shear force at every face of the surface (Fig 1D). By further integration, we calculated the force at every point along the centreline in a coordinate system attached to the larva's head (Fig 1E).

## Calculating bending moments

To calculate bending moments, we represented the fish by its central axis only. Effects of muscles, spine, and other tissues were combined for every transversal slice along this axis. This simplification allowed us to describe the fish as a nonlinear, one-dimensional beam in two-dimensional space.

We derived the equations of motion for this beam in an accelerating and rotating coordinate system attached to the fish's head (see S1 Text for details) [58]. We modelled the fish as a beam with varying cross sections undergoing arbitrarily large deformation. Plane cross sections are assumed to remain plane and perpendicular to the neutral line (no shear deformation), but axial deformation is allowed. In our tracking method, we assume that the fish deforms in a single plane (with an arbitrary three-dimensional position and orientation). Therefore, if we derive the equations in a reference frame attached to this deformation plane, we can use a two-dimensional beam model to represent the deformation of the fish. Because we chose to attach the noninertial reference frame to the head of the fish, the model contains both translation and rotational accelerations with respect to the inertial reference frame. We take these accelerations into account using fictitious forces [58].

In summary, we model the fish as a beam undergoing large bending deformations in two dimensions, leading to the following equations of motion expressed in a reference frame attached to the head of the fish (see section 1 in S1 Text for the derivation, and Fig S1 for definitions of the symbols):

$$\rho_{\text{fish}}A(s)\frac{\partial^2\xi(s,t)}{\partial t^2} = -\frac{\partial}{\partial s}(N(s,t)\cos\theta(s,t)) + \frac{\partial}{\partial s}(Q(s,t)\sin\theta(s,t))$$
$$+ f_{x,\text{muscle}}(s,t) + f_{x,\text{fluid}}(s,t) + f_{x,\text{fict}}(s,t)$$

(1)

$$\rho_{\text{fish}}A(s)\frac{\partial^2\eta(s,t)}{\partial t^2} = -\frac{\partial}{\partial s}(N(s,t)\sin\theta(s,t)) - \frac{\partial}{\partial s}(Q(s,t)\cos\theta(s,t))$$
$$+f_{y,\text{muscle}}(s,t)+f_{y,\text{fluid}}(s,t)+f_{y,\text{fict}}(s,t) \tag{2}$$

$$\rho_{\text{fish}}I(s)\frac{\partial^2\theta(s,t)}{\partial t^2} = -\frac{\partial M(s,t)}{\partial s} - Q(s,t) + m_{\text{muscle}}(s,t) + m_{\text{fluid}}(s,t) + m_{\text{fict}}(s,t) \tag{3}$$

where $s$ denotes the coordinate along the central axis of the fish; $t$ denotes time; $\rho_{\text{fish}}$ is the mass density of the fish; $A$ is the cross-sectional area distribution; $I$ is the second moment of area distribution; $\xi$ and $\eta$ are the displacement in, respectively, the $x$ and $y$ direction with respect to the undeformed configuration; $\theta$ is the deformation angle; $N$ is the normal force; $Q$ is the shear force; and $M$ is the bending moment.

In these equations, most terms were known from the three-dimensional surface model (the area and second moment of area distributions), the tracked video (the $x$ and $y$ displacement, deformation angle, and fictitious forces), and the CFD (the fluid forces and moment). The to-be-calculated unknown terms in the equations were the net normal forces, shear forces, and the bending moment. We described the distributions of these unknowns with quintic splines with uniformly spaced control points along the axis [56].

To determine the control point values of the normal force, shear force, and bending moment, we minimised the residuals of the equations of motion. For every trial combination of control points, we calculated the residuals of equations at all points along the fish. The squared sum of these normalised residuals was minimised with a Levenberg–Marquardt algorithm [57] to obtain the best-fitting control point values that meet the boundary conditions for both free ends (internal forces and moments are zero). When the residuals of the equations are equal to zero, the optimised distributions satisfy the governing equations and boundary conditions exactly. Therefore, this procedure ensured that the computed internal force and moment distributions (Fig 1F) were as close to physically valid as possible within the measurement error. From the motion of the centreline and fluid dynamic forces, we derived a local resultant power.

This optimisation procedure was validated with reference data obtained by integrating the equations of motion with a known external force distribution and internal moment distribution. We then reconstructed the bending moments, shear force, and normal force from the integrated motion and the prescribed external force distribution starting from different initial conditions, resulting in near-identical values (see section 5 in S1 Text).

The bending moment data were also used to calculate the resultant power from the fluid dynamic forces and the changes in kinetic energy. Note that we cannot calculate a meaningful value for the internal power, because we do not separate passive and active effects. The net power when combining these effects does not correspond to the actual power consumption of the fish, as the passive and active components might partly compensate each other.

## Comparing with a small-amplitude beam model

To compare our method to the previously used small-amplitude model [17,19,20], we reimplemented this model in Python, including periodic motion model, based on the original studies [17,31]. We reimplemented the periodic motion model from these studies, which is based on a truncated Fourier series. We fitted the Fourier coefficients to swimming motion with a least-squares method [59] to represent the motion of the reference swimmer as a series of Fourier coefficients. In addition, we reimplemented the bending moment reconstruction method from

the previous study [17] in Python. We manually digitised the amplitude and phase of displacement, curvature, and bending moment in their graphs (their Figs 4 and 5) and ran the reimplemented method on these data to verify that it gives near-identical results to the original method.

We applied the method to motion generated by integrating the beam equations with prescribed fluid force and muscle moment distributions (see S1 Text). We generated data sets for three different motion amplitudes by setting the amplitude of the muscle moment distribution to 2, 4, and 8 μN mm, resulting in peak-to-peak tail-beat amplitudes normalised by the fish lengths of 0.12, 0.20, and 0.35. First, we fitted Fourier coefficients (odd modes 1 through 9) to the lateral displacement, which we define as the displacement in $y$ direction (i.e., $\eta$ in our notation); the displacement in $x$ direction is not taken into account by the small-amplitude model. We also represent the lateral fluid force distribution as a series of Fourier coefficients; this is trivial because it is given by a pure cosine.

With the data in this form, we used the small-amplitude method to obtain the Fourier coefficients for the bending moment. To compare, we also applied our moment reconstruction method to the unfitted data.

## Calculating muscle cross-sectional area

We performed micro-computed-tomography (μCT) images of a 3-dpf zebrafish larva at the TOMCAT beamline at the Paul Scherrer Institut [60]. The larva was fixed in Bouin's solution and stained with phosphotungstic acid (PTA). The complete fish was imaged by stitching three scans with a resolution of $650 \times 650 \times 650$ nm per voxel. From these data, a centreline was extracted by finding the centre of area of each slice, segmented with simple thresholding. Finally, the muscle area was manually digitised in 51 planes, for which the image data were interpolated in a plane perpendicular to the centreline.

## Supporting information

**S1 Text. Mathematical background and validation.** In this supporting information, we provide the detailed mathematical background of the methods used in the article. In addition, we show the results of the validation performed on these methods. S1 Text contains five figures. Fig S1 illustrates the beam representation of the fish body for computation of bending moments. Fig S2 illustrates the reference data used for the internal forces and moments reconstruction. Fig S3 shows the comparison between reference and reconstructed internal forces and moments. Fig S4 shows the convergence of the immersed boundary solver. Finally, Fig S5 shows the comparison between the validated solver and the final choice of the settings of the immersed boundary solver.
(PDF)

**S1 Data. Underlying figure data.** In this supporting information, the underlying data for the following figures are provided: Fig 1G, Fig 3A and B, Fig 4E, Fig 5A–5D, Fig 6C–6H, Fig 7A–7E, Fig S4AB.
(XLSX)

## Acknowledgments

We thank the staff of the Carus animal facilities for providing the zebrafish larvae; Remco Pieters for building the camera setup; Stefan de Vilder from SDVISION for providing the pco. dimax HS4 camera; Herman ten Berge from Acal BFi Nederland for providing the Photron FASTCAM SA5 camera; Karen Léon-Kloosterziel, Remco Pieters, and Henk Schipper for their

assistance during the experiments; Wouter van Veen for helpful discussion on CFD; the Paul Scherrer Institut for use of the TOMCAT beamline at the Swiss Light Source, and Rajmund Mokso for providing guidance with the CT-scans.

## Author Contributions

**Conceptualization:** Cees J. Voesenek, Johan L. van Leeuwen.

**Data curation:** Cees J. Voesenek.

**Formal analysis:** Cees J. Voesenek, Johan L. van Leeuwen.

**Funding acquisition:** Johan L. van Leeuwen.

**Investigation:** Cees J. Voesenek, Gen Li, Florian T. Muijres, Johan L. van Leeuwen.

**Methodology:** Cees J. Voesenek, Gen Li, Florian T. Muijres, Johan L. van Leeuwen.

**Project administration:** Cees J. Voesenek, Johan L. van Leeuwen.

**Resources:** Johan L. van Leeuwen.

**Software:** Cees J. Voesenek, Gen Li, Johan L. van Leeuwen.

**Supervision:** Florian T. Muijres, Johan L. van Leeuwen.

**Validation:** Cees J. Voesenek, Gen Li, Johan L. van Leeuwen.

**Visualization:** Cees J. Voesenek.

**Writing – original draft:** Cees J. Voesenek, Johan L. van Leeuwen.

**Writing – review & editing:** Cees J. Voesenek, Gen Li, Florian T. Muijres, Johan L. van Leeuwen.

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
