## [Editor Report · Decision Letter 0]

26 Jun 2019

Dear Dr Van Leeuwen, 

Thank you very much for submitting your manuscript entitled "Fish larvae tackle the complex fluid-structure interactions of undulatory swimming with simple actuation" for consideration as a Research Article by PLOS Biology. 

We appreciated your patience while my colleagues and I have assessed your manuscript and consulted with two Academic Editors who are experts in this area. While we appreciate your work which developed a novel automated integrated experimental-numerical approach to study how fish larvae actuate their swimming motions in a simple and robust manner despite the involved complex physics and without a mature nervous system, our Academic Editors think, and we agree, that your study does not possess sufficiently novel biological insights which are required for consideration at PLOS Biology and that your method may lack generality and broader applicability and thus fits better in a more specialised journal.

While we cannot consider your manuscript for publication in PLOS Biology, we suggest that you consider transferring it to PLOS Computational Biology. The PLOS journals are editorially independent, so we cannot guarantee it will be reviewed there.

If you would like to transfer your manuscript, as suggested, please click the following link:

<DeepLinkData><DeepLinkTypeID>27</DeepLinkTypeID><peopleID>595760</peopleID><userSecurityID>2c80ae8e-373c-4900-87a7-62595f92f4a0</userSecurityID><documentID>36638</documentID><revision>0</revision><manuscriptNumber>PBIOLOGY-D-19-01779</manuscriptNumber><docSecurityID>45d93e1e-eda8-4c1f-ae9d-42a056c55eae</docSecurityID></DeepLinkData>

If you do NOT wish to transfer your manuscript, please click this link to decline: 

<DeepLinkData><DeepLinkTypeID>28</DeepLinkTypeID><peopleID>595760</peopleID><userSecurityID>2c80ae8e-373c-4900-87a7-62595f92f4a0</userSecurityID><documentID>36638</documentID><revision>0</revision><manuscriptNumber>PBIOLOGY-D-19-01779</manuscriptNumber><docSecurityID>45d93e1e-eda8-4c1f-ae9d-42a056c55eae</docSecurityID></DeepLinkData>

Please note, you can log into the submission sites with the same login that you used to submit to this journal. 

Should you choose to transfer your submission you will receive a confirmation email within 24-48 hours after accepting the transfer. If you have any questions, please feel free to contact the journal at plosbiology@plos.org.

Thank you again for your interest in PLOS Biology. 

Sincerely,

Di Jiang, PhD

Associate Editor

PLOS Biology

---

## [Decision Letter · Decision Letter 1]

25 Sep 2019

Dear Dr Van Leeuwen,

Thank you very much for submitting your manuscript "Fish larvae tackle the complex fluid-structure interactions of undulatory swimming with simple actuation" for consideration as a Initial Research Submission at PLOS Biology. Your manuscript has been evaluated by the PLOS Biology editors, an Academic Editor with relevant expertise, and by three independent reviewers.

In light of the reviews (below), we will not be able to accept the current version of the manuscript, but we would welcome resubmission of a much-revised version that addresses all of the three reviewers' detailed and constructive comments on the manuscript. You will need to make the paper suitable for our Methods and Resources format, to address the question of what new value your sophisticated modelling approach adds, and to provide all of the information and code needed for others to make use of your method. Our academic editor advises that reviewer 3’s comments on the framing of the manuscript are especially apposite in this respect and that you will need to pay particular regard to these and to the related comments from reviewer 2. We emphasise that our decision on any revision will rest entirely on the extent to which it meets the specific criteria for a Methods and Resources paper, which will require a substantial reframing of the paper. We cannot make any decision about publication until we have seen the revised manuscript and your response to the reviewers' comments. Your revised manuscript is also likely to be sent for further evaluation by the reviewers.

Your revisions should address the specific points made by each reviewer. Please submit a file detailing your responses to the editorial requests and a point-by-point response to all of the reviewers' comments that indicates the changes you have made to the manuscript. In addition to a clean copy of the manuscript, please upload a 'track-changes' version of your manuscript that specifies the edits made. This should be uploaded as a "Related" file type. You should also cite any additional relevant literature that has been published since the original submission and mention any additional citations in your response. 

Before you revise your manuscript, please review the following PLOS policy and formatting requirements checklist PDF: http://journals.plos.org/plosbiology/s/file?id=9411/plos-biology-formatting-checklist.pdf. It is helpful if you format your revision according to our requirements - should your paper subsequently be accepted, this will save time at the acceptance stage.

Please note that as a condition of publication PLOS' data policy (http://journals.plos.org/plosbiology/s/data-availability) requires that you make available all data used to draw the conclusions arrived at in your manuscript. If you have not already done so, you must include any data used in your manuscript either in appropriate repositories, within the body of the manuscript, or as supporting information (N.B. this includes any numerical values that were used to generate graphs, histograms etc.). For an example see here: http://www.plosbiology.org/article/info%3Adoi%2F10.1371%2Fjournal.pbio.1001908#s5.

For manuscripts submitted on or after 1st July 2019, we require the original, uncropped and minimally adjusted images supporting all blot and gel results reported in an article's figures or Supporting Information files. We will require these files before a manuscript can be accepted so please prepare them now, if you have not already uploaded them. Please carefully read our guidelines for how to prepare and upload this data: https://journals.plos.org/plosbiology/s/figures#loc-blot-and-gel-reporting-requirements.

Upon resubmission, the editors will assess your revision and if the editors and Academic Editor feel that the revised manuscript remains appropriate for the journal, we will send the manuscript for re-review. We aim to consult the same Academic Editor and reviewers for revised manuscripts but may consult others if needed.

We expect to receive your revised manuscript within two months. Please email us (plosbiology@plos.org) to discuss this if you have any questions or concerns, or would like to request an extension. At this stage, your manuscript remains formally under active consideration at our journal; please notify us by email if you do not wish to submit a revision and instead wish to pursue publication elsewhere, so that we may end consideration of the manuscript at PLOS Biology.

When you are ready to submit a revised version of your manuscript, please go to https://www.editorialmanager.com/pbiology/ and log in as an Author. Click the link labelled 'Submissions Needing Revision' where you will find your submission record. 

Sincerely,

Di Jiang

PLOS Biology

Reviewer remarks:

Reviewer #1: The authors investigate the actuation of fish larvae for undulatory swimming. They develop a model for the dynamics of undulatory swimming and use tracking data from experiments and CFD to estimate the external hydrodynamic forces and deduce the distribution of internal bending moments. Similar inverse dynamics approaches have been used in previous studies, and are cited appropriately. The approach presented here is detailed and robust and supports the analysis of a large dataset. The study yields new interesting insight into the actuation dynamics. I only have a few comments.

1./ The authors define “vigour”. The definition seems a bit arbitrary and I don’t think the vigour physically represents anything, or else it should be explained. The effort on the otherhand, seems to be the more physical quantity, and relates to a maximum power input (also units of effort in fig 5 are given in Newtons, but should be in units of Watts). Why do the authors not use effort or the mean resultant power in their analysis on figure 4. 

2./ Related to point 1./. Fig 3 c.d. seem to suggest some sort of bimodal distribution of larvae between larvae that accelerate strongly and swim slowly and larvae that swim faster but don’t accelerate so strongly. It seems that the definition of vigour collapses the two artificially, but these modes should be different in their actuation. While these two modes may be similar in the spatial distribution of the actuation (fig.4 cd), they may be different in the time distribution of the actuation (fig. 4ef only represents the phase of maximum bending). These differences may be characterized in fig 2 c.d., by looking at the time distribution of curvature and moment.

3./ The supplemental information contains most of the useful information, to understand the approach. It could benefit from minor reorganization, as reading through was at times confusing because of the sequence.

Reviewer #2: This study is certainly impressive and thorough methodologically. My comments therefore focus on the broader scientific messages, most of which should be simple to deal with in revision.

1. Is the hypothesis as expressed worthwhile? Or is it actually almost circular? The hypothesis (L.14) can be summarized as: a simple brain results in simple actuation. Is this not a (sorry) no-brainer? If it is the brain that initiates activation, is it not part of the definition of a simple brain that it should only be capable of simple activation?

2. Is the result notable or surprising? (And does it matter if it is not?) L. 78-84. That amplitude and frequency determines things sounds pretty intuitive. You may wish to expand on why this is not the case, or whether it does not matter that it is intuitive. In effect, your finding could be interpreted as evidence of no change in ‘gait’. Similarly, a ‘trot’ can be used by a horse across a range of speeds and accelerations with only changes in frequency and amplitude. Note that I am NOT saying that your findings are not interesting (after all, why ‘should’ a fish stick to one ‘gait’?). But a small addition or two might prevent the casual reader from being immediately dismissive. 

3. Does this observation actually require the new methodology? Would (or have) similar conclusions been arrived at from a much simpler kinematic study, or much simpler fluid modeling? Yes, the case is made that previous studies were deficient in some aspect (small/large deflection beam theory, intermediate Re fluids etc.)… but the same can (and indeed you do) be said to some extent about the current study. Inasmuch as all studies and models are deficient in some way, can the case be made for why this one in particular is importantly less deficient? What findings were directly due to the improved methods? What false findings were avoided?

4. I suggest being more explicit and earlier about the motivation behind using customized derived metrics (effort and vigor). To what extent is this with the purpose of ‘collapsing variation’ (L.141) – so effectively being a Principal Component with defined units. And, if a parameter is very highly correlated with power (and, given the units, is this surprising?) why not stick to using power? To some extent I feel Figure 3 to be a demonstration that the relationships between force, work, power, and between hydrodynamic and whole-body… all sort of relate intuitively.

Minor and line comments follow.

There appears to be the implication of an adaptive slant... and this does not feel justified. (L.22 allows function during development). 

The suggestion that complex physics would be (initially) thought to require a sophisticate control system (L. 41) probably overstates matters. Most biologists should be familiar with complex physics occurring with very simple (or zero) control.

To what extent is the lack of curvature towards the tail tip a consequence of the shape reconstruction? I am not sure how this could be dealt with neatly… but I am suspicious that a 90 degree bend in the last 1% might get smoothed out, whereas the same angle bend at 50% would make for an obviously right-angle fish, and would persist. I don’t think this affects the story of the paper, but if it is an inevitable consequence of methodology and not a reliable measurement, this should be noted.

Reviewer #3: See attached file.

---

## [Decision Letter · Decision Letter 2]

31 Jan 2020

Dear Dr Van Leeuwen,

Thank you very much for submitting a revised version of your manuscript "Experimental-numerical method for calculating bending moments in swimming fish shows that fish larvae control undulatory swimming with simple actuation" for consideration as a Methods and Resources at PLOS Biology. This revised version of your manuscript has been evaluated by the PLOS Biology editors, the Academic Editor and the original reviewers.

In light of the reviews (below), we will not be able to accept the current version of the manuscript, but we would welcome re-submission of further revised version that takes into account the reviewers' comments. You will need to address all the points raised by the reviewers. Our Academic Editor wishes to emphasise the need to compare with previous methods and to make further effort to reframe your study as a Methods and Resources paper in your revision. We would also like to stress reviewer 3's concern regarding a possible circularity in your use of your "c parameter". Furthermore, you will need to make your code available. We cannot make any decision about publication until we have seen the revised manuscript and your response to the reviewers' comments. Your revised manuscript is also likely to be sent for further evaluation by the reviewers.

We expect to receive your revised manuscript within 2 months. 

Sincerely,

Di Jiang

PLOS Biology

REVIEWS:

Reviewer #1: The authors have addressed the comments satisfactorily. 

Reviewer #2: This revision reads nicely and appears appropriate as a Methods and Resources Article.

My single major comment is that perhaps a new methodology should be compared with the older ones. Fine, the case is made that it is bound to be better… but how different, and when are the differences most significant? And which of the additions make the difference?

If we could start off with the 'oldest' of small-angle beam theory, simplified 2-d and steady-state, could some quantification be made of the changes due to the 'newest' of large-deflection beams, modern 3-d CFD, unsteady? Even if it was just looking at the extreme cases of biggest deflection etc…. just how different (let us assume wrong) would the old methods give?

Very minor comments

I presume you are avoiding discussion of various 'efficiency' terms for a reason… it probably falls outside the scope of this paper; if not, around Line 145 might be a place to comment.

Line 178. Reynolds numbers aren't actually given from what I can see (merely that they fall in an 'intermediate' regime). But we are thinking that Re is high enough for drag (or resultant force) to be broadly proportional to V^2 not V, aren't we?

Passage 375-386 felt a little like a revisiting of something already stated…?

---

## [Decision Letter · Decision Letter 3]

13 May 2020

Dear Dr Van Leeuwen,

Thank you for submitting your revised Methods and Resources entitled "Experimental-numerical method for calculating bending moments in swimming fish shows that fish larvae control undulatory swimming with simple actuation" for publication in PLOS Biology. I have now obtained advice from two of the original reviewers and have discussed their comments with the Academic Editor. 

Based on the reviews, we will probably accept this manuscript for publication. However, we would like you to consider moving the new Figure S4 into the main text. Given that this is a Methods and Resources manuscript, the new Figure S4 comparing the results of the new large-amplitude model to the small-amplitude standard and high-fidelity CFD is especially valuable and may be worth being elevated to a main figure. The Figure S4 will probably stand alone in the main text without the need to move too much else from supplementary; but we would leave the final decision on this to you. 

We expect to receive your revised manuscript within two weeks. Your revisions should address the specific points made by each reviewer. In addition to the remaining revisions and before we will be able to formally accept your manuscript and consider it "in press", we also need to ensure that your article conforms to our guidelines. A member of our team will be in touch shortly with a set of requests. As we can't proceed until these requirements are met, your swift response will help prevent delays to publication.

*Copyediting*

*Published Peer Review History*

*Early Version*

*Submitting Your Revision*

Sincerely,

Di Jiang, PhD

Associate Editor

PLOS Biology

ETHICS STATEMENT:

-- Please create a separate Ethics Statement subsection in the beginning of the Methods section and please include the animal protocol number. For details, please see below. 

-- Please include the full name of the IACUC/ethics committee that reviewed and approved the animal care and use protocol/permit/project license. Please also include an approval number.

-- Please include the specific national or international regulations/guidelines to which your animal care and use protocol adhered. Please note that institutional or accreditation organization guidelines (such as AAALAC) do not meet this requirement.

-- Please include information about the form of consent (written/oral) given for research involving human participants. All research involving human participants must have been approved by the authors' Institutional Review Board (IRB) or an equivalent committee, and all clinical investigation must have been conducted according to the principles expressed in the Declaration of Helsinki.

DATA POLICY:

-- Regardless of the method selected, please ensure that you provide the individual numerical values that underlie the summary data displayed in the following figure panels as they are essential for readers to assess your analysis and to reproduce it: current figures 1B-G, 2B-H, 3A-D, 4A-H, 5A-E, S2CD, S3A-I, S4A-F, S5AB, S6AB, S7A-D. NOTE: the numerical data provided should include all replicates AND the way in which the plotted mean and errors were derived (it should not present only the mean/average values).

-- Please provide a reviewer/editor key/token for https://doi.org/10.5061/dryad.2280gb5p6 so we can check the data before acceptance. 

Reviewer remarks:

Reviewer #2: I am grateful for the considerable work put in to responding to my previous comments.

The current draft looks very nice. Any wider adoption of the new methods may end being dependent on how easily they can be implemented using the code being made available. 

Reviewer #3 (Eric D Tytell, signed review): The authors have sufficiently addressed my concerns and the manuscript is ready for publication.

---

## [Editor Report · Decision Letter 4]

30 Jun 2020

Dear Dr Van Leeuwen,

On behalf of my colleagues and the Academic Editor, Graham K Taylor, I am pleased to inform you that we will be delighted to publish your Methods and Resources in PLOS Biology. 

Early Version

PRESS 

Kind regards,

Vita Usova

Publication Assistant, 

PLOS Biology

on behalf of

Di Jiang, PhD,

Senior Editor

PLOS Biology